# Estimation of Off-Target Dicamba Damage on Soybean Using UAV Imagery and Deep Learning

**DOI:** 10.3390/s23063241

**Published:** 2023-03-19

**Authors:** Fengkai Tian, Caio Canella Vieira, Jing Zhou, Jianfeng Zhou, Pengyin Chen

**Affiliations:** 1Department of Biomedical, Biological and Chemical Engineering, University of Missouri, Columbia, MO 65211, USA; 2Crop, Soil, and Environmental Sciences, Bumpers College, University of Arkansas, Fayetteville, AR 72701, USA; 3Biological Systems Engineering, University of Wisconsin-Madison, Madison, WI 53706, USA; 4Division of Plant Science and Technology, University of Missouri, Columbia, MO 65211, USA

**Keywords:** soybean, dicamba tolerance, high-throughput phenotyping, deep learning

## Abstract

Weeds can cause significant yield losses and will continue to be a problem for agricultural production due to climate change. Dicamba is widely used to control weeds in monocot crops, especially genetically engineered dicamba-tolerant (DT) dicot crops, such as soybean and cotton, which has resulted in severe off-target dicamba exposure and substantial yield losses to non-tolerant crops. There is a strong demand for non-genetically engineered DT soybeans through conventional breeding selection. Public breeding programs have identified genetic resources that confer greater tolerance to off-target dicamba damage in soybeans. Efficient and high throughput phenotyping tools can facilitate the collection of a large number of accurate crop traits to improve the breeding efficiency. This study aimed to evaluate unmanned aerial vehicle (UAV) imagery and deep-learning-based data analytic methods to quantify off-target dicamba damage in genetically diverse soybean genotypes. In this research, a total of 463 soybean genotypes were planted in five different fields (different soil types) with prolonged exposure to off-target dicamba in 2020 and 2021. Crop damage due to off-target dicamba was assessed by breeders using a 1–5 scale with a 0.5 increment, which was further classified into three classes, i.e., susceptible (≥3.5), moderate (2.0 to 3.0), and tolerant (≤1.5). A UAV platform equipped with a red-green-blue (RGB) camera was used to collect images on the same days. Collected images were stitched to generate orthomosaic images for each field, and soybean plots were manually segmented from the orthomosaic images. Deep learning models, including dense convolutional neural network-121 (DenseNet121), residual neural network-50 (ResNet50), visual geometry group-16 (VGG16), and Depthwise Separable Convolutions (Xception), were developed to quantify crop damage levels. Results show that the DenseNet121 had the best performance in classifying damage with an accuracy of 82%. The 95% binomial proportion confidence interval showed a range of accuracy from 79% to 84% (*p*-value ≤ 0.01). In addition, no extreme misclassifications (i.e., misclassification between tolerant and susceptible soybeans) were observed. The results are promising since soybean breeding programs typically aim to identify those genotypes with ‘extreme’ phenotypes (e.g., the top 10% of highly tolerant genotypes). This study demonstrates that UAV imagery and deep learning have great potential to high-throughput quantify soybean damage due to off-target dicamba and improve the efficiency of crop breeding programs in selecting soybean genotypes with desired traits.

## 1. Introduction

Soybean [*Glycine max* (L.) Merr.] is a globally important and widely used leguminous crop for food, feed, and biofuel production due to its high seed protein and oil content traits [1]. In the United States, soybean acreage is projected to reach 87 million acres by 2023 [2], and global soybean trade in terms of soybean is expected to increase by 22% in 2025 [3]. However, soybean production is limited by various adverse abiotic and biotic factors, such as extreme environmental conditions (e.g., drought and flooding), pests and pathogens, and weeds [4,5,6,7]. Weeds are suppressive invasive species that can significantly reduce the soybean yield and seed quality [8]. Commonly used integrated weed management practices include the post-emergent control of weeds, such as the application of herbicides [9] and mechanical approaches [10]. Herbicides are widely used in soybean production due to their high efficiency and relatively low cost.

Dicamba, a chlorobenzene acid herbicide, is one of the most widely used herbicides in soybean production to control a broad spectrum of woody plants and broadleaf weeds [11]. The global dicamba herbicide market is expected to be more than $10 billion by 2030 from $4.5 billion in 2020 [12]. Based on market research data and aggregated sales data, about half of dicamba-tolerant (DT) cotton and soybean fields applied dicamba more than one time in the U.S. in 2020 [13]. However, dicamba is highly volatile and can move to off-target crops and plants via particle and vapor drift in a large region (i.e., off-target dicamba). Off-target dicamba can cause severe injury and significant yield loss to non-tolerant crops. It was reported that dicamba-injured soybeans had 3.6 million acres in 2017 [14]. Therefore, developing soybean genotypes naturally tolerant to dicamba is important for improving soybean production.

Conventional soybean breeding programs develop new varieties through extensive field trials and phenotypic and genotypic screening to select genotypes with high-yielding potential and desirable economic traits, including off-target dicamba tolerance [15]. However, the assessment of off-target dicamba damage requires breeders to check hundreds or thousands of genotypes across multiple environments (combination of location, field, and year), which is time-consuming and labor-intensive [16]. In addition, visual scores may not be accurate due to their subjective nature [16]. In recent years, unmanned aerial vehicle (UAV) imaging technology has been extensively used in agriculture for collecting the high spatiotemporal resolution imagery data of crops [17]. Thanks to the advance in data analytic technologies and computer vision, various deep convolutional neural networks (CNNs) have been employed to address a range of agricultural challenges. For instance, researchers have applied pre-trained CNNs to classify soybean leaf diseases due to environments [18], estimate soybean leaf defoliation via synthetic images and AlexNet [19], predict the soybean maturity date using a CNN-LSTM model [20], forecast soybean yield through a mixed CNN model [21], and assess stand count using Resnet18 [22].

In our previous study, machine learning methods were tested to classify off-target dicamba damage using UAV imagery with an accuracy of 0.75 [16]. However, machine-learning approaches require the manual extraction of image features, which takes time to process and may potentially introduce more human bias [23]. The manual pre-processing procedures may make it challenging to handle a large and complex dataset efficiently and accurately. On the other hand, deep learning uses multiple layers of neurons consisting of complex structures or non-linear transformations, which has been used for processing imagery data with complexity [24]. Deep learning models have been used to process and analyze large-scale and complicated imagery data [25], including UAV-based crop phenotyping data [26].

This study aimed to evaluate the soybean tolerance to off-target dicamba exposure using UAV imagery and deep learning techniques. There were two supporting objectives, including (1) identifying suitable deep learning models for assessing off-target dicamba damage, and (2) analyzing the effects of environmental factors and data collection time on the model performance. The study also compared the performance of different pre-trained models and discussed the limitations and potential biases of the study. It is expected that the developed methods in this study could be used to improve the efficiency and accuracy of soybean breeding programs.

## 2. Materials and Methods

### 2.1. Field Experiment and Plant Materials

This study was conducted at the University of Missouri Fisher Delta Research, Extension, and Education Center in Portageville, MO, United States (36°24′51.7″ N, 89°41′58.3″ W) in 2020 and 2021. The elevation of the fields is approximately 85 m above sea level, and the average temperature was 28.3 °C in the typical soybean growing season from early May to late September in 2020, and 27.0 °C in 2021. The 2020 experiment consisted of 230 advanced genotypes derived from 115 unique bi-parental populations and ten commercial cultivars as checks, including seven DT and three glyphosate-tolerant (GT) cultivars. In 2021, the experiment consisted of 209 advanced genotypes derived from 103 different bi-parental populations and 15 commercial checks (11 DT and 4 GT cultivars). Field trials were conducted using a three-replicated randomized complete block design in different environments, as listed in Table 1. In 2020 and 2021, different genotypes were planted as 4-row plots (four 3.66 m rows spaced 0.76 m) in seven fields, resulting in a total of 3731 soybean plots (including checks). Table 1 lists all the data collection times, which were determined based on the early reproductive stages (R1 to R3, approximately 100 to 130 DAP) [27].

### 2.2. Field Evaluation of Dicamba Damage

This study evaluated a total of 2300 plots from five different fields in 2020 and 1431 plots from two different fields in 2021 (details in Table 1). Each tested genotype had approximately three replicates, and 16 of 463 commercial cultivars had nine replicates. Typical visual symptoms of dicamba damage include the crinkling and cupping of newly developing leaves, reduced canopy coverage, and plant stunting. Soybean damage was scored on a 1–5 scale with 0.5 increments following the criteria described by Vieira, Sarkar [16], i.e., 1—no to minimal visual symptoms, 2—limited cupping of the newly-developing leaves and no visual impact on the canopy coverage and vegetative growth, 3—accentuated cupping of the newly-developing leaves and moderate reduction in the canopy area and vegetative growth, 4—severe cupping of the newly-developing leaves and a pronounced reduction in the canopy area and vegetative growth, and 5—extreme damage with severe cupping of the newly-developing leaves and intense reduction in the canopy coverage and vegetative growth. In this study, soybean genotypes were classified into three tolerance classes according to their rated damage scores, including “tolerant” (1 and 1.5), “moderate” (2, 2.5, and 3), and “susceptible” (3.5, 4, and 4.5). Figure 1 illustrates examples of soybean genotypes at different tolerance classes to off-dicamba damage.

### 2.3. Image Data Collection

Imagery data were acquired using a UAV platform (DJI Phantom 4 Pro, DJI, Shenzhen, China) with a built-in red-green-blue (RGB) camera at a resolution of 5472 × 3648 pixels. Images were taken at 1 frame per 2 s at 20 m above ground level (AGL) and a flying speed of 7 km h^−1^ in both years. The UAV platform was configured to follow a zigzag path with an overlap of 70% and a side overlap of 65% using the flight control software Autopilot (Hangar Technology, Austin, TX, USA). Each image frame was geo-referenced by the UAV’s onboard global navigation satellite system (GNSS). The image ground sampling distance (GSD) was 5.5 mm per pixel. Data were collected around noon with sufficient sunshine under a clear sky condition.

### 2.4. Image Processing

Orthomosaic images for each field were generated using the software Agisoft Metashape 1.8.4 (Agisoft, St. Petersburg, Russia) following the standard protocol provided by the software. Each orthomosaic image was exported in “*tiff*” (Tag Image File Format) format and processed using the Computer Vision System Toolbox and Image Processing Toolbox of MATLAB (ver. 2021a, MathWorks, Natick, MA, USA) to segment all plots of each field. Background information (shadow, soil, and plant residues) was removed by applying a threshold (pixel value ≤ 200) to the gray images converted from RGB images using the “*rgb2gray*” function in MATLAB. The processed images included individual plots with only the soybean canopy and were ready to be used for developing models. The image size of each plot was adjusted to the size of 224 × 224 pixels (required by all models) using the function “*resize*” from the package TensorFlow (ver. 2.8.0) and the method of interpolation was set to be “nearest” so each pixel value would be rounded as an integer. In addition, the hue from the HSV color scale was calculated using the function “*bgr2hsv*” from the package OpenCV (version 4.5.5) in Python.

### 2.5. Damage Assessment Using Deep Learning Model

Several pre-trained convolutional neural network (CNN) deep learning models were tested to classify the soybean genotypes into three defined off-target dicamba tolerance classes (i.e., tolerant, moderate, and susceptible). The visual geometry group (VGG) is a popular CNN architecture with multiple convolutional layers and VGG16 has been used to classify soybean leaf diseases, weed pressure, and crop stress [18,28,29]. Due to the good general performance, VGG16 was tested in this study to quantify soybean tolerance to off-target dicamba. The second CNN model tested is the residual neural network-50 (ResNet50) which has been used to successfully assess the soybean leaf defoliation and corn emergence [19,30]. In addition, the study also evaluated the Depthwise Separable Convolutions (Xception) that adopted the depthwise separable convolution and was used for crop growth prediction and crop disease recognition [31,32]. Xception splits the computation into depthwise convolution (a single convolutional filter per input channel) and pointwise convolution (a linear combination of the output of the depthwise convolution), which makes it faster and requires fewer computing resources compared to standard convolution. The last tested model was the Dense Convolutional Network-121 (DenseNet121), which directly connects any layer to all subsequent layers. The DenseNet121 has been used to classify the damage levels of cotton aphids and nutrient deficiencies in rice [33,34]. All models were implemented in the Keras platform (version 2.10.0), an open-source application programming interface (API) for deep learning written in Python (version 3.9.7).

Figure 2 illustrates the architecture of a DenseNet121 model, where the RGB images of each plot were entered as input, followed by different conventional layers, and four dense blocks with three transition layers. After the last dense block, global average pooling 2D was added to perform downsampling, followed by a dense layer. The *softmax* activation function calculated a probability for each possible class. The “*argmax*” function from the package NumPy (Version 1.21.5) was used to retain a class that contains the highest possibility for each image. The loss function was defined as the sparse categorical cross-entropy. The grid search process was performed using the function “*GridSearchCV*” in the scikit-learn library [35]. The parameters and their searching scope were: batch_size: [16, 32, 64, 100, 128], learning_rate: [0.001, 0.05, 0.01], and optimizer: [SGD, Adam, nAdam]. An ideal number of training epochs was determined when the testing loss stopped decreasing [36]. Each class was assigned a weight (*w_i_*) for balancing the loss function during the training process, as defined in Equation (1).
(1)$$w_{i}=\frac{1}{n_{i}}\times \frac{N}{3}\;$$
where, *n_i_* is the number of samples in each (*i* = 1, 2, 3) of the three tolerance classes, and *N* is the total number of samples from all three classes.

Other pre-trained CNN models, including VGG16, ResNet50, and Xception, were implemented using similar architecture settings as Figure 2. According to Ref. [37], the last few layers in the pre-train deep learning models are typically use-specific and need to be fine-tuned according to the new tasks. Therefore, in this study, only the dense layer or convolutional blocks were set to be trainable, followed by the output of three soybean tolerance classes. Other layers of each pre-trained model were frozen and their weights were not updated by the optimizer during training processing in order to reduce the risk of overfitting [38]. Different models have the same options for tuning parameters in order to compare them at the same level. All models were trained and tested using the same dataset with an 80:20 split [39] for training and testing the datasets.

### 2.6. Classification Metrics

All models were implemented in R (Version 4.1.2). The model performance was assessed using the metrics of *accuracy*, *precision*, *recall*, *95% confidence interval (CI)*, and *F*1-*score* defined by Equations (2)–(6). The *accuracy* is the ratio of the number of correct predictions divided by the total number of predictions of the models. In this study, *precision* and *recall* were used to represent the percentage of true positives out of the predicted positive samples and the percentage of true positives out of the true positive samples. The *F*-*score* is defined as the harmonic mean of recall and precision.
(2)$$Accuracy=\frac{\left(TP+TN\right)}{\left(TP+FP+FN+TN\right)}$$
(3)$$Precision=\frac{TP}{TP+FP}$$
(4)$$Recall=\frac{TP}{TP+FN}$$
(5)$$F1-score=2\times \frac{Precision\times Recall\;}{Precision+Recall}$$
(6)$$95\%\;CI=\hat{p}\;\pm z\sqrt{\frac{\hat{p}\left(1-\hat{p}\right)}{n}}$$
where, *TP* is True Positive that a test result correctly indicates the true event. *TN* is True Negative, where a test result correctly indicates the false event. *FP* is False Positive, where a test result falsely indicates the true event. *FN* is False Negative, where a test result falsely indicates the true event. The n is the number of samples and $\hat{p}\;$is the probability of success.

The *p*-value with a significance level of 0.01 is used to determine the statistical significance of a model using the one-sided binomial test from the “caret” package in R. The 95% binomial proportion confidence interval demonstrates that there is a 95% level of confidence that the actual value of accuracy falls within the specified range.

### 2.7. Statistical Analysis

A one-way analysis of variance (ANOVA) was performed to evaluate whether there was a significant difference in a particular image feature between the years 2020 and 2021. The “*aov*” function of R Studio was used to perform the ANOVA analysis. A significant difference was considered for the *p*-value ≤ 0.01 obtained from the hypothetical test.

## 3. Results and Discussion

### 3.1. Distribution of Visual Dicamba Damage Score

The distribution of off-dicamba damage scores of all plots evaluated by breeders is shown in Figure 3. The figure shows that the largest number of plots were classified in the moderate class, accounting for 75.9% (1746 plots) in 2020 and 68.7% (983 plots) in 2021. The number of susceptible plots accounted for 13.4% (309 plots) in 2020 and 9.2% (132 plots) in 2021, and the number of tolerant plots accounted for 10.7% (245 plots) in 2020 and 22.1% (316 plots) in 2021. The distribution of plot numbers in three categories was aligned with the breeder’s expectation that most soybean genotypes have a moderate response to off-target dicamba damage, which is consistent with the findings of [40]. Conventional breeding programs usually select 10% of genotypes to be advanced throughout the pipeline based on their observations. The results show that the observed number of tolerant genotypes was more than 20% of all test genotypes in 2021, which was more than double that observed in 2020. According to Ref. [40], the observed trend of higher tolerance to off-target dicamba damage among the genotypes could be explained by the indirect selection process employed in conventional breeding pipelines. Conventional breeding programs prioritized various breeding trials with favorable agronomic traits and higher yields in environments with prolonged exposure to dicamba. Moreover, the phenotypes of soybean plots could also be influenced by various environmental factors, as discussed in the following section.

### 3.2. Influence of Environmental Factors

This study was conducted in seven fields over two consecutive years, resulting in a wide variation in the plant growth environments. In addition to off-target dicamba injury, environmental variation has a significant impact on the plant characteristics. Figure 4 shows the selected examples of UAV images captured for soybean genotypes at each response category in different environments. The figure provides visual evidence of the variation in the plant characteristics (color and canopy cover) in response to off-target dicamba in different environments. In general, plant genotypes of all classes in 2021 had greater canopy coverage and greener color than those in 2020, which could be due to the different amount of off-target dicamba on the genotypes, the environmental conditions (weather and soil type), and the time of damage exposure and assessment. One extreme example is Field 4 (2020), where the tolerant and moderate genotypes have less canopy cover than the susceptible plots. The results suggest that breeders assessed the damage scores by considering multiple factors that may be influenced by the seed emergence, seed quality, and disease [41]. Many soybean plots under a similar environment expressed various phenotypes. Previous research also found that among all the variables, the image feature “*Hue*” had the greatest impact on the classification accuracy of the off-target dicamba damage using a random forest model [16]. The results of the one-way ANOVA test showed a statistically significant difference in “*Hue*” between 2020 and 2021 (*p*-value ≤ 0.01), which further demonstrated the dissimilarity between the two years of the dataset.

In conventional breeding programs, breeders assess the off-target dicamba damage by visiting each plot and assigning a score accordingly based on their experience. Visual scores are typically subjective and can be affected by environmental factors, such as light conditions and growth stages. In addition, the evaluation process is time-consuming and labor-intensive. Therefore, there is a pressing need to develop a high-throughput crop phenotyping system to accurately (objectively) and efficiently evaluate the tolerance of soybean genotypes.

### 3.3. Performance Evaluation of Classification Models

The distribution of different response classes (as shown in Figure 3) indicates that the sample numbers of the three classes are unbalanced, which might affect the model’s performance [42]. To improve the model performance, a weight for each class was assigned based on Equation (1) to mitigate the effects of unbalanced data. In this study, the weight for the susceptible, moderate, and tolerant classes was 2.74, 0.46, and 2.165, respectively.

The model was evaluated using a uniform test dataset consisting of 746 images, including 95 susceptible (12.7%), 527 moderate (70.6%), and 124 tolerant genotypes (16.6%). The overall performance (2-year data) of the test models was reported in Table 2, including the classification accuracy, precision, recall, and F1-score. The table shows that all tested models achieved a competitive or higher classification accuracy than our previous study (0.75) [16]. DenseNet121 achieved the highest classification accuracy of 0.82, followed by ResNet50 (0.80), VGG16 (0.76), and Xception (0.75). The results also reveal that the *p*-values for all four models were lower than 0.01, indicating the effectiveness of the model performance.

Table 3 shows the confusion matrix of the four deep learning models in the classification of off-target dicamba tolerance, which is used to calculate the Precision, Recall, and F1-Score (in Table 2) using Equations (2)–(6). Based on Table 2, different models showed significantly better performance for a given class and metric. For example, ResNet50 achieved the highest precision of 0.77 in classifying the susceptible class, and the greatest precision for the class tolerant is 0.73 from DenseNet121. Moreover, ResNet50 could successfully identify 76% (0.76) of the positive samples (TP + FP) in the tolerant class and DenseNet121 had the highest recall for the susceptible class. However, the overall performance of DenseNet121 and ResNet50 outperformed other models (VGG16 and Xception) in discriminating between the tolerant and susceptible genotypes. In addition, according to Table 3, DenseNet121 was the only model that did not misclassify the tolerant plots as susceptible, and vice-versa. Although ResNet50 was better than DenseNet121 in some metrics, DenseNet121 achieved the best F1-scores of 0.59, 0.87, and 0.72 in the susceptible, moderate, and tolerant classes, respectively. The F1-scores were better than those obtained using traditional machine learning methods, which were 0.36, 0.83, and 0.68 for the three responding classes [16]. In addition, the accuracy of DenseNet121 achieved the highest accuracy of 0.82 with 95% confidence intervals between 0.79 and 0.84. The performance of the test models in this study showed a similar conclusion from Shazia, Xuan [43], who also found that the DenseNet121 was better than ResNet50, VGG-16, and Xception in the classification of the X-ray images.

### 3.4. Model Performance Using Single-Year Dataset

This part of the study aimed to compare the results of the training and test datasets on the two-year datasets separately versus training on a combined two-year dataset to determine whether the latter can provide a better result or not. Only the best overall performance model, DenseNet121, was evaluated. The model was configured using the same parameters as described in the previous sections. The test dataset for the data of 2020 consisted of 460 images, of which 10.2% were visually classified as susceptible, 77.4% as moderate, and 12.4% as tolerant. The 2021 test dataset consisted of 286 samples with 8.0% susceptible, 70.6% moderate, and 13.3% tolerant genotypes. Table 4 and Table 5 show the results for each year. The results show that DenseNet121 achieved similar classification accuracy using all two-year data or single-year data, with slightly higher accuracy in the year 2020. We noticed that the accuracy for the year 2020 was slightly higher than the results reported in Section 3.3. This is mainly due to the higher density of the moderate class in the 2020 dataset, which has a positive impact on the accuracy because the results for the moderate class consistently outperformed other classes. On the other hand, F1-values for the classification of susceptible and tolerant were actually lower (0.52 and 0.71, respectively). In contrast, training the model on datasets from both years could achieve 0.59 and 0.72, which was also higher than the F1-values for 2021. Furthermore, the “estimate_sigma” function from the Skimage library in Python was used to assess the robustness of the wavelet-based estimator of the Gaussian noise standard deviation. The results indicate that the test dataset in this study had little noise. Therefore, the image noise was considered non-significant in affecting the model performance. In summary, the DenseNet121 showed better performance in 2020 compared to 2021, which could be due to the greater dataset in 2020 than that in 2021, or due to the differences in the soybean phenotypes and environmental factors between the two years. The use of a combined dataset from both years allows models to be trained using data from a broad range of environmental conditions, which may improve the model robustness for quantifying the soybean tolerance to off-target dicamba.

### 3.5. Research Limitations

One of the limitations of this study is that the amount of dicamba applied to soybean was not controlled. Since the off-target dicamba exposure may occur at different growth stages at different dosages, often with an unknown source of exposure, it is usually hard to quantify the herbicide exposure for each plot [44]. The lack of control over dicamba might affect the accuracy of the findings. Another limitation is the absence of controlled groups. Different soybean genotypes might exhibit various phenotypic characteristics, and their development can vary from time to time [45]. A lack of control groups may have caused bias in determining the damage levels due to off-target dicamba. Moreover, manual segmentation of each soybean plot from orthomosaic images was time-consuming and labor-intensive. Although there are available tools for automatic segmentation of the orthomosaic images, such as FIELDimageR [46], they may not work efficiently in this case due to the significant variation in the size of the soybean canopy caused by the dicamba damage.

## 4. Conclusions and Future Work

The developed UAV-based phenotyping pipeline aimed to quantify the soybean response to off-target dicamba faster and more accurately than visual observations. Using machine learning and one-year data, it achieved an accuracy of 0.75 in distinguishing tolerant, moderate, and susceptible soybean genotypes from off-target dicamba damage [16]. This study evaluated the performance of four pre-trained deep learning models in classifying the soybean tolerance to off-target dicamba using a two-year dataset. Results show that the deep learning model DenseNet121 achieved the highest classification accuracy of 0.82 in differentiating the soybean genotypes from three tolerance classes. The tested deep learning models outperformed machine learning models, although the two-year dataset used for training deep learning models included a large variation in the environmental conditions (different years and soil conditions) and more than 400 genotypes. Meanwhile, the selected deep learning model DenseNet121 also performed better using single-year datasets than machine learning methods. In addition, compared to machine learning methods, deep learning models do not need additional feature extraction steps, which are time-consuming and may introduce human bias. Therefore, the proposed deep learning methods and UAV imagery have great potential for selecting tolerant soybean genotypes to off-target dicamba.

More experiments and data are needed to improve the overall performance of deep learning models in quantifying the soybean tolerance to off-target dicamba. In addition, more advanced deep learning models will be tested, such as adversarial discriminative domain adaptation [47] or the graph convolutional network [48], which are expected to improve accuracy and robustness in the future. Meanwhile, the amount of dicamba exposure of each field or plot needs to be quantified, and the same should be done for the environmental factors, such as the soil conditions. It is also worth testing other types of sensors, including multispectral, hyperspectral, and thermal imagers, that may acquire additional crop traits and that quantify the genetic variation of crops due to environmental stresses.

## Figures and Tables

**Figure 1 sensors-23-03241-f001:**

Visual assessment of three tolerance classes of soybeans to off-target dicamba damage, i.e., (**a**) tolerant, (**b**) moderate, and (**c**) susceptible.

**Figure 2 sensors-23-03241-f002:**
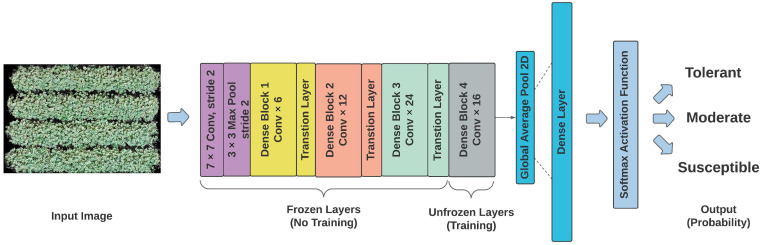
Illustration of the DenseNet121 model architecture.

**Figure 3 sensors-23-03241-f003:**
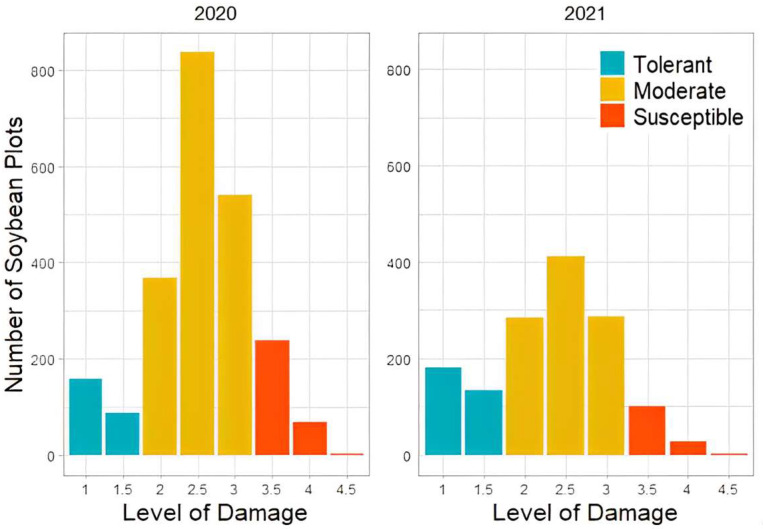
Distribution of off-target dicamba damage scores assessed by breeders in 2020 and 2021. The horizontal axis has the visual scores that were classified into three tolerance classes, i.e., Tolerant (blue), Moderate (yellow), and Susceptible (red).

**Figure 4 sensors-23-03241-f004:**
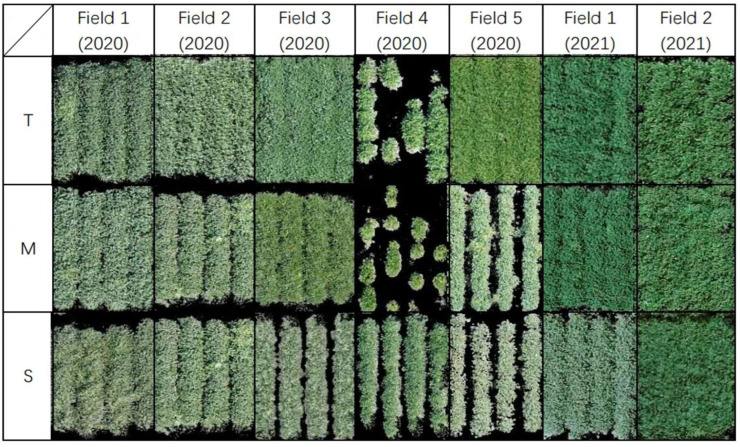
Examples of UAV images of different soybean genotypes at three tolerance classes to off-target dicamba damage. Images were selected from genotypes in different fields in different years to illustrate the variation in plant characteristics. T: tolerant, M: moderate, S: susceptible.

**Table 1 sensors-23-03241-t001:** Summary of field experiments and data collection in 2020 and 2021.

Year	Field	Genotype Number	4-Row Plots ^1^	Planting Date	Field Assessment and Imaging Data	DAP ^2^
2020	1	213	670	17 April 2020	20 August 2020	125
	2	213	670	28 April 2020	8 September 2020	134
	3	213	672	18 April 2020	21 August 2020	125
	4	48	144	1 June 2020	15 September 2020	105
	5	48	144	27 May 2020	14 September 2020	110
2021	6	223	714	22 April 2021	12 August 2021	112
	7	223	717	17 May 2021	16 August 2021	91

^1^ Include checks. ^2^ DAP, days after planting, refers to the number of days between the time of field evaluation and planting.

**Table 2 sensors-23-03241-t002:** Summary of measurable metrics for the selected deep learning models in the classification of off-target dicamba tolerance.

	DenseNet121			ResNet50			VGG16			Xception		
	S *	M *	T *	S	M	T	S	M	T	S	M	T
*Precision*	0.66	0.85	0.73	0.77	0.84	0.64	0.56	0.85	0.58	0.51	0.84	0.60
*Recall*	0.54	0.89	0.71	0.39	0.88	0.76	0.53	0.81	0.71	0.49	0.81	0.72
*F*1-*score*	0.59	0.87	0.72	0.51	0.86	0.69	0.55	0.83	0.64	0.50	0.82	0.66
*Accuracy*		0.82			0.80			0.76			0.75	

* S: Susceptible, M: Moderate, T: Tolerant.

**Table 3 sensors-23-03241-t003:** Confusion matrix of the selected four deep learning models for the classification of off-target dicamba tolerance.

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

S	M	T	S	M	T
*Precision*	0.60	0.88	0.85	0.54	0.81	0.84
*Recall*	0.47	0.94	0.61	0.30	0.94	0.54
*F*1-*score*	0.52	0.91	0.71	0.39	0.87	0.66
*Accuracy*	0.85	0.80

**Table 5 sensors-23-03241-t005:** Confusion matrix of DenseNet121 in years 2020 and 2021.

		Reference—2021			Reference—2021		
		S	M	T	S	M	T
Predict	S	22	15	0	7	6	0
	M	25	335	22	16	190	28
	T	0	6	35	0	6	33

## Data Availability

The data presented in this study are available on request from the corresponding author.

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
