# Peer review of "Estimation of Off-Target Dicamba Damage on Soybean Using UAV Imagery and Deep Learning"

_sensors, 2023, doi:10.3390/s23063241_

Round 1
Reviewer 1 Report
Accepted
Author Response
Dear Reviewer,
We appreciate your feedback and comments on improving the manuscript quality. It is our honor to have your support. Meanwhile, we tried our best to improve the structure, language, and grammar in this revision.
Thanks,
Authors
Reviewer 2 Report
This manuscript proposed a novel method for estimating off-target dicamba damage on soybean, where deep learning and UAV imagery were used for the task of interest. In the proposed method, dense convolutional neural network-121 (DenseNet121), residual neural network-50 (ResNet50), visual geometry group-16 (VGG16), and Depthwise Separable Convolutions (Xception) were developed to quantify crop damage levels. Finally, the experimental verification was conduced to prove the effectiveness of the proposed technique, with promising results. Overall, the topic of this research is interesting, and the structure of manuscript was well configured. I suggest that it needs a minor revision by addressing the following comments, before the manuscript is accepted for publication.
1. The main novelty and contributions of this investigation is suggested to be clearly clarified in abstract and introduction.
2. Broaden and update literature review on CNN and its application in data/image processing. E.g. Vision-based concrete crack detection using a hybrid framework considering noise effect; Torsional capacity evaluation of RC beams using an improved bird swarm algorithm optimised 2D convolutional neural network
3. In this study, DensNet, ResNet, VGG and Xception are all belong to pre-trained deep CNNs, which were developed for other tasks. However, the authors used them for crop damage detection in this paper. Hence, transfer learning technique was employed. Please add more information about how these models were transferred.
4. The performance of CNNs are related to the hyperparameter setting. How did the authors set the optimal network parameters to achieve the best identification accuracy?
5. Training time is suggested as a metric for model performance evaluation.
6. How about the robustness of the proposed method against noise effect in images?
7. More future research should be included in conclusion part.
Author Response
Dear Reviewer,
We highly appreciate your feedback and comments on improving the manuscript quality. We tried our best to address all your valuable comments and improve the structure, language, and grammar by carefully proof-reading the revision. We have revised the manuscript accordingly and the revised content is highlighted in blue color. Our responses to your specific comments are as follows:
- The main novelty and contributions of this investigation is suggested to be clearly clarified in abstract and introduction.
Response: We clarified the main novelty and contributions of this paper in lines 36 – 39, 91-92, & 95 – 96 in the revised manuscript.
- Broaden and update literature review on CNN and its application in data/image processing. E.g. Vision-based concrete crack detection using a hybrid framework considering noise effect; Torsional capacity evaluation of RC beams using an improved bird swarm algorithm optimised 2D convolutional neural network
Response: More recent literature regarding CNN and its applications for crop analysis was added in lines 73 -80.
- In this study, DensNet, ResNet, VGG and Xception are all belong to pre-trained deep CNNs, which were developed for other tasks. However, the authors used them for crop damage detection in this paper. Hence, transfer learning technique was employed. Please add more information about how these models were transferred.
Response: Pertinent information was included in lines 203 - 208 to clarify the adaption process of pre-trained deep CNNs for our specific tasks.
- The performance of CNNs are related to the hyperparameter setting. How did the authors set the optimal network parameters to achieve the best identification accuracy?
Response: The optimal network parameters were decided by grid search. More information about grid search has been added from lines 191 - 193.
- Training time is suggested as a metric for model performance evaluation.
Response: Thanks for your suggestion. However, we did not consider the training time as an informative metric in our case. All the test models have a relatively small group of trainable parameters, and the training process could be completed within 3 minutes in the Google Colab platform [CPU: Intel Xeon @2.2GHz, 6 cores, and GPU: Nvidia A-100-SXM4, 40GB]. Therefore, the training time may not be able to reflect their difference in model performance. We will compare their differences in our future study when we have sufficient datasets.
- How about the robustness of the proposed method against noise effect in images?
Response: Pertinent information was added in lines 351 - 355 to explain the low level of image noise in our image dataset.
-  More future research should be included in conclusion part.
Response: We added more information about future research in lines 396-402.
Reviewer 3 Report
Please see attached file.

Author Response
Dear Reviewer,
We highly appreciate your feedback and comments on improving the manuscript quality. We tried our best to address all your valuable comments and improve the structure, language, and grammar by carefully proof-reading the revision. We have revised the manuscript accordingly and the revised content is highlighted in blue color. Our responses to your specific comments are as follows:
- The main contributions and motivations for the present work have to be clearer.
Response: We clarified the main novelty and contributions of this paper in lines 36 – 39, 91-92, & 95 – 96 in the revised manuscript.
- The processed images included individual plots with only soybean canopy and were ready to be used for developing models. Did the authors tried with other if no mention one line about the issues faced.
Response: We have added more on the research limitations to reflect the current issues that we have faced during the imaging processing. Pertinent information was added in lines 372 - 377.
- Figure 2. The Architecture of the selected DenseNet121 model is good but should be more informative.
Response: We added an additional explanation in Figure 2 (Line 200).
- To validate the model's performance, a consistent train (80%) – test (20%) split was used for all models. Why authors choose 80 and 20 why not 70 and 30?
Response: Based on our experiences in using these models, we found both 70/30 and 80/20 split rations are reasonable, which also is confirmed by other studies [1]. In addition, the 80/20 training testing split is widely used and recommended by [2-3]. Hence, we used 80/20 split ratio in our study. Pertinent information was added in lines 210 - 211.
- Figure 3. Distribution of off-target dicamba damage scores assessed by breeders in 2020 and 2021. The horizontal axis has the visual scores that were classified into three tolerance classes, i.e., Susceptible (red), Moderate (yellow), and Tolerant (blue). The legends are not clear make a clear representation.
Response: We revised the legend of Figure 3 (253). We deleted the legend title since the color can clearly indicate each class and rearranged the legend to display three classes in a top-to-bottom order.
- The results of the one-way ANOVA test showed a statistically significant difference in “Hue” between 2020 and 2021 (p-value≤0.01), which further proves the dissimilarity between the two years of the dataset. The authors should explain the reason of optimal performance in detail.
Response: Thanks for your suggestion. The ANOVA test was used to evaluate the variations of plant characteristics (e.g., color, canopy cover) in response to off-target dicamba in various environments. The large variations due to environmental factors make it challenging for breeders' decision-making for selections. The information justified the need for using high-throughput crop phenotyping technology as developed in this study.
Reference:
[1] Gholamy, A., V. Kreinovich, and O. Kosheleva, Why 70/30 or 80/20 relation between training and testing sets: A pedagogical explanation. 2018.
[2] Training and Test Sets: Splitting Data. Available from: https://developers.google.com/machine-learning/crash-course/training-and-test-sets/splitting-data.
[3] The 80/20 Split Intuition and an Alternative Split Method. Available from: https://towardsdatascience.com/finally-why-we-use-an-80-20-split-for-training-and-test-data-plus-an-alternative-method-oh-yes-edc77e96295d.